# Training load quantification of high intensity exercises: Discrepancies between original and alternative methods

**François-Denis Desgorces**[1,2]*, **Jean-Christophe Hourcade**[1,3], **Romain Dubois**[4], **Jean-François Toussaint**[1,2,5], **Philippe Noirez**[1,2]

**1** Institut de Recherche bioMédicale et d'Épidemiologie du Sport, Institut National du Sport de l'Expertise et de la Performance, Paris, France, **2** Université de Paris, Équipe d'accueil, Paris, France, **3** Académie Conseil en Préparation Athlétique (ACPA Performance), Gradignan, France, **4** Mouvement, Équilibre, Performance, Santé, Équipe d'Accueil, Université de Pau et des Pays de l'Adour, Tarbes, France, **5** Centre d'Investigations en Médecine du Sport, Hôpital Hôtel-Dieu, Assistance Publique-Hôpitaux de Paris, Paris, France

* francois.desgorces@unilim.fr

**Data Availability Statement:** All relevant data are within the manuscript and its Supporting Information files.

## Abstract

The purpose of this study was to quantify training loads (TL) of high intensity sessions through original methods (TRIMP; session-RPE; Work-Endurance-Recovery) and their updated alternatives (TRIMP$_{cumulative}$; RPE$_{alone}$; New-WER). Ten endurance athletes were requested to perform five sessions until exhaustion. Session 1 composed by a 800m maximal performance and four intermittent sessions performed at the 800m velocity, three sessions with 400m of interval length and work:recovery ratios of 2:1, 1:1 and 1:2 and one with 200m intervals and 1:1. Total TL were quantified from the sessions' beginning to the cool-down period and an intermediate TL (TL$_{800}$) was calculated when 800m running was accumulated within the sessions. At the end of the sessions high and similar RPE were reported (effect size, $\eta^2 = 0.12$), while, at the intermediate 800m distance, the higher interval distances and work:recovery ratios the higher the RPE ($\eta^2 = 0.88$). Our results show marked differences in sessions' total TL between original (*e.g.*, lowest TL for the 800m and highest for the 200m-1:1 sessions) and alternative methods (RPE$_{alone}$ and New-WER; similar TL for each session). Differences appear in TL$_{800}$ notably between TRIMP and other methods which are negatively correlated. All TL report light to moderate correlations between original methods and their alternatives, original methods are strongly correlated together, as observed for alternative methods. Differences in TL quantification between original and alternative methods underline that they are not interchangeable. Because of high exercise volume influence, original methods markedly enhance TL of sessions with higher exercise volumes although these presented the easiest interval distances and work-recovery ratios. Alternative methods based on exhaustion level (New-WER) and exertion (RPE$_{alone}$) provided a new and promising point of view of TL quantification where exhaustion determines the highest TL whatever the exercise. This remains to be tested with more extended populations submitted to wider ranges of exercises.

**Funding:** The authors received no specific funding for this work.

**Competing interests:** The authors have declared that no competing interests exist.

## Introduction

Forty five years ago, analyzing the relations between training and performance Banister et al. (1975) defined "training load" (TL) as the key parameter for measuring some "dose" of exercise, or effort induced by training [1]. On the basis of their original work and of several papers published thereafter, we assume that TL is the exercise-induced physiological strain resulting from the combination of exercise's intensity, volume and density influences [2–4]. Most original TL quantification methods multiply exercise volume by an indicator of exercise intensity.

The main exercise parameters for training programs are as follow: *i)* intensity; *ii)* volume (distance of one single interval and total accumulated distance); *iii)* density caused by recovery period duration (frequently expressed through work: recovery ratio) and sometimes by the intensity of the recovery time [5]. These parameters are interdependent. Any change of parameters may have some incidence on physiological strain on the condition that the other parameters remain constant. Within the context of an accurate training description and of a comparative analysis of exercise's effects, the method used to assess TL must allow for an accurate quantification of all exercises regardless of the parameters. However, some authors consider that exercises with higher volumes should result in higher TL based on their expected major effect on performance. Unfortunately, this approach leads to predefining some of the exercises' TL prior to the analysis of the dose-response relationship. We posit that TL should remain a measure of the exercise dose without considering its expected effects.

Based on their comparative study of TL quantification methods, some authors suggested that no method can qualify as a "gold standard" today [4,6]. The training impulse (TRIMP) method, proposed by Banister et al. is based on heart-rate (HR) records for intensity assessment. It is widely accepted as the historical method that raised awareness about the need for training quantification and monitoring athletes in a new way [1,4]. Although TRIMP is deemed as a valuable tool to quantify continuous and prolonged exercises, some authors pointed out that the use of mean HR could fail in reflecting the physiological demand of intermittent exercises, thus supporting the idea of original method adaptations [7,8]. HR records may be impractical in a training context. On the account of the relations between the rating of perceived exertion (RPE) and HR, some authors therefore suggested the use of RPE instead of HR to assess the intensity of the session for TL quantification [9]. The session-RPE (S-RPE) method of multiplying the session duration by RPE has thus been proposed twenty-five years ago and, since then, it has been widely used by scientists and coaches [10,11]. However, several authors suggested that RPE itself may provide an accurate assessment of TL which did not require any multiplication by the exercise duration [6,12]. This idea was reinforced by studies describing the noticeable influence of exercise duration on RPE [13,14]. Based on the "accumulated work: work limit" and "work: recovery" ratios, the work endurance recovery method (WER) has been developed to enhance the accuracy of TL quantification of intermittent exercises in regard to TRIMP and S-RPE [3]. Although WER has been adopted and used for training programs of combat and ball game sports [15,16], Hourcade et al. (2018) have reported unwanted variations in WER TL when intensity distribution was changed within sessions [17]. This study suggested that WER should be based on the achievable endurance limit of each exercise session. The assessment of that limit must be aligned with the structure of the session (*e.g.* interval distance, work:recovery ratio. . .).

Therefore, some methodologists have argued that original methods need to be improved in an attempt to define alternative ones. Nevertheless, the choice of any of the quantification methods appears to be based on their usability rather than their ability to assess the "dose of effort" [4,10]. The metrics capacity used with different methods to provide equated TL for

varied exercises is of a matter of importance in sport sciences although this issue may require further analysis.

This study set sessions performed until exhaustion: one 800m performance session and four high intensity sessions varying in interval distance and work-recovery ratios performed at the same 800m specific velocity. Our main purpose was to analyze the sessions' TL that has been quantified through both original and alternative methods. Potential discrepancies in TL quantifications, and their origins, were then discussed.

## Materials and methods

### Subjects

Ten recreational middle and long distance male runners (19.2±1.2 years; height and weight respectively 178.1±4.9 cm and 67.9±7.5 kg) participated in the study. Based on their performance level during the study period, subjects were deemed in good mental and physical condition, and declared not to be under any medication. All subjects were trained for 6 to 10 hours per week in the last two years of our study. Their training comprised of a minimum of one high intensity interval session and two long distance running sessions. The remaining time in their training schedule was allocated to either running or other sport activities. The study period lasted three weeks allowing for five exercise sessions to be performed. This study followed the guidelines of the Declaration of Helsinki, and was approved by the French Sport Sciences Ethics Committee. All subjects provided their written informed consent prior to their enrollment.

### Design

Each session was performed at 10.00 a.m. on an outdoor athletics track. Special attention was paid to consistent air temperature and hygrometry conditions (respectively, 14.6 ± 2.1˚C and 47.9 ± 3.3). The study protocol increased the number of exhausting sessions at high intensity, thus changing subjects' training habits. Subjects were asked to arrive for each session in a rested state, and suspend their regular training at least 24h prior to each session and avoid exhausting exercises in the study period.

Each session started with the same standardized 20min warm-up and ended with a standardized cool-down period. The warm-up was a 10min run at low-moderate velocity, followed by a 4min series of specific exercises (athletic drills, balance, strengthening) and three 100m runs at the 800-m specific velocity (interspersed by a 1min recovery after each run). The cool-down consisted in a 4min passive recovery period followed by a 6min run within the 60–65% of the $HR_{max}$ zone. Each session ended after this cool-down routine.

The goal for the first session was to perform a maximal 800m race. Subjects were then randomly assigned to complete four interval sessions in a random order, with a minimum three-day period between each session (3.5 ± 0.6 days). As in a maximal 800m race, each interval session was performed until exhaustion. Here exhaustion designates the inability of a subject to maintain the expected velocity during two 100m intervals in a row. The 800m performance was used to determine the specific velocity that subjects had to maintain throughout the interval sessions. In doing so, the exercise intensity remained unchanged throughout the study. During 400m runs, individual velocity was determined by a beep from the runner's watch at each 100m interval (50m in 200m runs). For each participant, one 200m interval length session with a work-to-rest ratio of 1:1 was performed (based on a duration index). Three 400m interval length sessions were then performed with ratios of 2:1, 1:1, 1:2. Recovery times were meant to be passive as they involved slow walks and static positions. To illustrate this, Fig 1 shows how 800m and 400m-1:2 sessions were conducted.

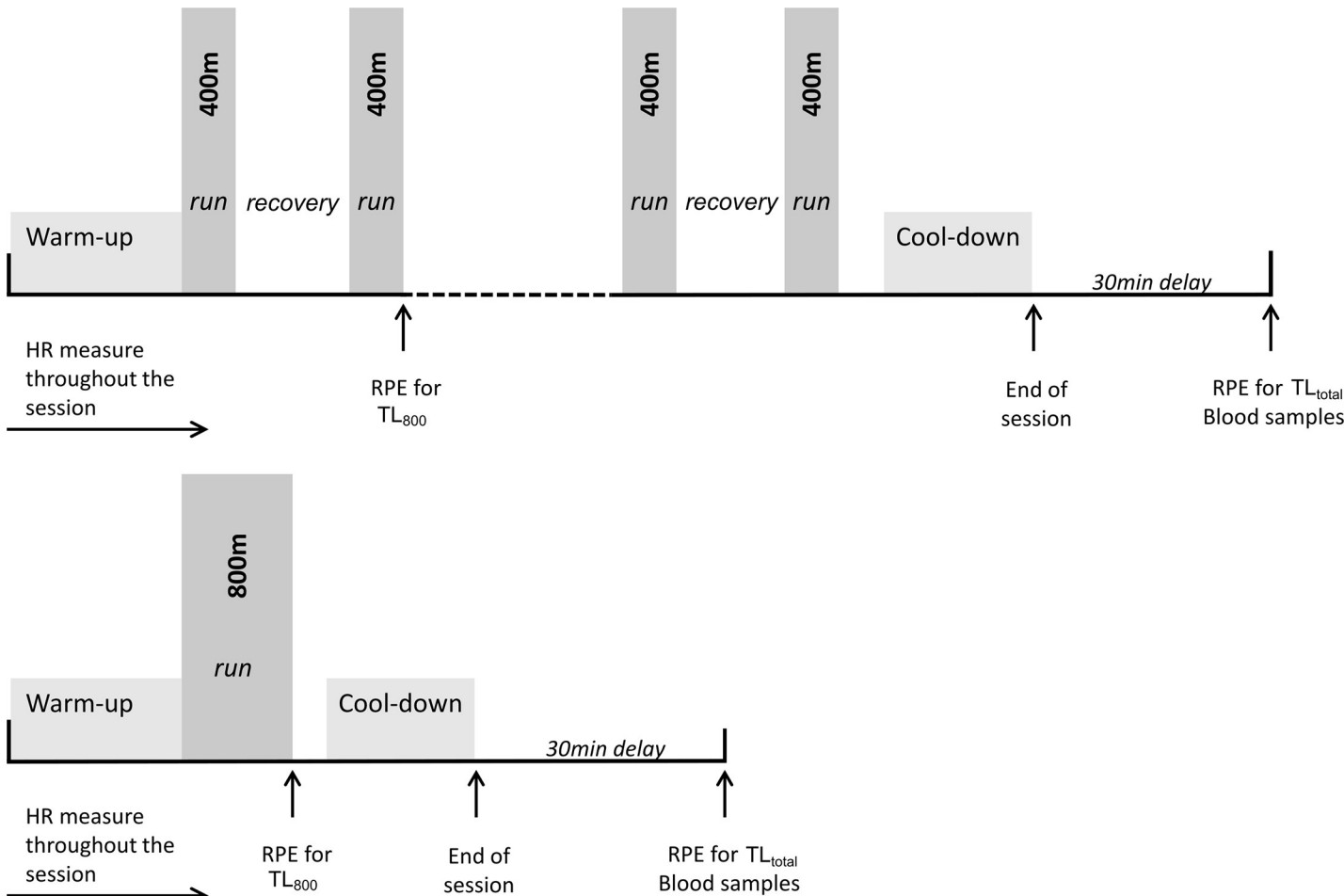

**Fig 1. Protocol of 800m and 400m-1:2 sessions.** Sessions were composed by warm-up and cool-down and by a 800m performance or by 400m intervals with recovery periods (two times longer than work duration) performed until exhaustion.

## Training load quantification

TL was quantified using three original methods (TRIMP, S-RPE and WER) and their respective updated methods: cumulative TRIMP ($TRIMP_c$), RPE alone ($RPE_{alone}$) and new WER method (NeWER). $TL_{total}$ was calculated for the entire session (from warm-up to cool down). For the purpose of this study, an intermediate TL ($TL_{800}$) was calculated for all sessions from the warm-up until the completion of 800m of run at the 800m performance velocity.

TRIMP was determined by Banister et al. (1975), as "Eq 1":

$$TRIMP = TD \times HRr \times 0.64 \times e(1.92 \times HRr)$$

TD in Eq 1 refers to the training duration of the effective training session expressed in minutes. HRr is the HR reserve determined from Eq 2:

$$HRr = \frac{(HRts - HRb)}{HRmax - HRb}$$

$HR_{ts}$ in Eq 2 designates the average training session HR. $HR_{max}$ and $HR_b$ are the maximal and basal HR, respectively.

$TRIMP_c$ was chosen as TRIMP updated method because it enabled to improve quantification of intermittent exercises, requiring fewer adaptations than with other TRIMP alternative methods [8,18]. In brief, partial TL for exercise intervals and recovery periods were calculated on the basis of the TRIMP method ($TRIMP_c$ being the summation of all partial TL) [8].

TL was also quantified using the S-RPE method proposed by Foster et al. [19]. Athletes' RPE assessment was multiplied by the entire duration (in minutes) of their training session. As an alternative, basic RPE ($RPE_{alone}$) was also regarded as a valid TL assessment method [10,12].

TL was also calculated using the WER method as previously described by using Eq 3 [3]:

$$TL = \left(\frac{accumulatedWork}{Endlim}\right) + \ln\left(1 + \frac{DurationaccumulatedWork}{Durationaccumulatedrecovery}\right)$$

The accumulated work for any given exercise session represented the total amount of work that had been completed at the requested intensity while $End_{lim}$ was the endurance limit at such intensity. Although inducing a low TL level, warm-up and cool-down periods were also quantified and added to the TL of the assigned exercise. An exponential regression analysis of individual running performances (800m, 1,500m and 10,000m or 21,000m) was developed to estimate the endurance limit corresponding to warm-up and cool-down intensity levels. The passive recovery times during the cool-down and warm-up periods could not be quantified.

NeWER quantified sessions' TL by using the ratio of the session's accumulated exercise duration against the individual maximum duration in the considered exercise construction. NeWER is expressed as a percentage of exhaustion for the considered session. In a 400m-1:1, for instance, a four-minute running time may represent 50% of the maximum running sequence (eight minutes). The same method was used to calculate TL for warm-up and cool-down periods. These three TL values were added to the whole exercise TL. In brief, a best-fitting method using exponential models was applied to the exercise HR and to the maximal exercise durations that were recorded during the 400m-2:1, 400m-1:1 and 200m-1:1 sessions. This was done to determine maximal durations at HR levels recorded during the warm-up and cool-down periods.

## Measurements

HR was measured and recorded every 5 seconds with individually coded HR transmitters to avoid interference (Polar RS 400, Polar Electro, Kempele, Finland) during each session. The highest HR reached during the exercise sessions or the 800m performance was defined as the individual $HR_{max}$. In the first week of the study, $HR_b$ was self-measured in the supine position during a 3-min period each morning for three days. The lower of the three values was then chosen as the individual $HR_b$. As mentioned above, sessions' mean HR and the total time spent in the 95 to 100% of $HR_{max}$ zone were registered to define the sessions' goal of soliciting oxygen consumption [20,21].

Capillary blood samples were drawn from one finger pulp within 3min following each training session to be analyzed with the new Lactate pro II analyzer (Arkray, Kyoto, Japan).

RPE was obtained using Foster et al. modified category ratio scale (CR-10) [19]. Two weeks prior to the study, subjects were taught how to use the CR-10 scale during their training sessions. RPE assessments during the sessions were conducted by asking participants "How much exertion are you feeling right now?" showing them the CR-10 scale with verbal anchors (from rest: 0, to maximal exercise: 10). Compliant with the S-RPE quantification method, $TL_{total}$ was based on the RPE that was recorded 30min after ending each session, including the 800m maximal performance session. To calculate $TL_{800}$, however, RPE was also recorded after 800m were accumulated during the recovery period, as the entire session was not yet finished, and right after the 800m performance.

## Statistical analysis

Statistical analysis was undertaken using R software (version 3.6.2) (R Foundation for Statistical Computing, Vienna, Austria). After assessing variance homogeneity with a Levene test, TL and measures to each session (HR, blood lactate, exercise volumes) were compared using the linear mixed-models approach (nlme package), which is a one way ANOVA with repeated measures (TL × session) and (response × session). Data normality was assessed graphically (see S1 Fig). Whenever any statistical difference was detected, adjusted Bonferroni post-hoc analyses only were conducted with simultaneous tests for general linear hypotheses (Emmeans package). When variances were found to be heterogeneous, a specific non-parametric randomized block analysis of variance was used: the Friedman test. Post-hoc analyses were then conducted with the Wilcoxon-Nemenyi-McDonald-Thompson test (coin and multcomp packages) using Tal Galili's code, published on r-statistics.com (https://www.r-statistics.com/2010/02/post-hoc-analysis-for-friedmans-test-r-code). In both cases, the one way ANOVA and Friedman test effect sizes were calculated. Correlations between TL (both $TL_{total}$ and $TL_{800}$ quantifications), or between TL and assessed parameters were analyzed using the Pearson moment correlation. Confidence intervals for Pearson correlation coefficients were also calculated (psychometric package). Mean and standard deviations (mean ± SD) were calculated for each parameter and statistical significance was set at $p < 0.05$.

## Results

Mean 800m performance was 129.6 ± 6.7 s (22.3 km.h$^{-1}$). Subjects' $HR_{max}$ and $HR_b$ were respectively 196 ± 4.6 and 44.8 ± 6.2 bpm. The 20min standardized warm-up led to a mean HR of 135.2 ± 5.2 bpm without any difference between the sessions (p = 0.9) as for cool-down HR (128.8 ± 5.9 bpm, p = 0.6).

Table 1 reports the total sessions' duration, HR measures, blood lactate concentrations and effort perception at the end of the sessions and after 800m had been accumulated. Effort perception and running distance in cool-down did not differ between the sessions (ANOVA respectively, p = 0.2 and p = 0.07). Mean HR of sessions presented few discrepancies, but was

**Table 1. Records of exercise durations, heart-rate (HR), blood lactate concentrations, distance covered in the 6min run during the cool-down period and ratings of perceived exertion (CR-10); *i)* at the end of the sessions conducted until exhaustion (see top of the table); and *ii)* when 800m of run were accumulated (see bottom of the table).** Results of parameter variance analysis from the ANOVA and Friedman test, and calculated effect size ($\eta^2$). Results with different manuscript letters (a, b, c, d, e) are significantly different (p<0.05; one way-Anova and Tukey test).

| *Records at exhaustion* | 400m-2:1 | | 400m-1:1 | | 400m-1:2 | | 200m-1:1 | | 800-m | | *Variance* | $\eta^2$ |
|---|---|---|---|---|---|---|---|---|---|---|---|---|
| **Session total duration (min)** | 33.7 | ± 0.6[a] | 37.3 | ± 1.0[b] | 45.8 | ± 1.9[c] | 44.5 | ± 1.3[c] | 32.2 | ± 0.2[a] | *p = 0.0003* | *0.97* |
| **Exercise total distance (m)** | 920 | ± 123[a] | 1360 | ± 183[b] | 1950 | ± 184[c] | 2690 | ± 197[d] | 800 | ± 0[e] | *p<0.0001* | *0.99* |
| **Session mean HR (bpm)** | 138.8 | ± 4.1[a] | 142.1 | ± 4.5[b] | 142.4 | ± 4.0[b] | 144.9 | ± 4.7[b] | 135.8 | ± 3.9[a] | *p<0.0001* | *0.53* |
| **Exercise mean HR (bpm)** | 182.7 | ± 6.1[a] | 177.7 | ± 3.9[a] | 166.4 | ± 4.0[b] | 180.7 | ± 4.8[a] | 190.5 | ± 4.6[c] | *p<0.001* | *0.65* |
| **Time in 95–100% HR zone (s)** | 126.4 | ± 12.3[a] | 209.2 | ± 25.4[b] | 324.6 | ± 49.5[c] | 501.2 | ± 77.1[d] | 114.8 | ± 5.9[e] | *p<0.0001* | *0.98* |
| **Blood lactate (mmol.l-1)** | 18.2 | ± 2.6[a] | 15.2 | ± 2.1[b] | 11.6 | ± 2.3[c] | 13.6 | ± 2.4[d] | 18.1 | ± 2.4[a] | *p<0.0001* | *0.58* |
| **Distance run in cool down (m)** | 1102 | ± 40 | 1157 | ± 38 | 1131 | ± 43 | 1109 | ± 39 | 1124 | ± 45 | *p = 0.07* | *0.18* |
| **CR-10 after 30 min** | 9.30 | ± 0.5 | 9.70 | ± 0.5 | 8.90 | ± 1.1 | 9.30 | ± 0.9 | 9.50 | ± 0.5 | *p = 0.21* | *0.12* |
| *Records stopped at 800-m of run* | | | | | | | | | | | | |
| **Session duration (min)** | 23.2 | ± 0.2[a] | 24.3 | ± 0.2[a] | 26.5 | ± 0.3[b] | 24.3 | ± 0.2[a] | 22.2 | ± 0.1[c] | *p = 0.01* | *0.40* |
| **Session mean HR (bpm)** | 141.5 | ± 5.0 | 142.3 | ± 5.1 | 141.3 | ± 4.9 | 142.5 | ± 5.2 | 138.7 | ± 4.8 | *p = 0.05* | *0.05* |
| **Exercise mean HR (bpm)** | 181.2 | ± 3.7[a] | 175.1 | ± 4.8[b] | 160.3 | ± 4.3[c] | 176.7 | ± 5.9[b] | 190.5 | ± 4.6[d] | *p<0.0001* | *0.83* |
| **Time in 95–100% HR zone (s)** | 123.7 | ± 15.7[a] | 126.2 | ± 15.5[a] | 114.7 | ± 17.6[a] | 143.8 | ± 16.7[b] | 114.8 | ± 5.9[a] | *p = 0.005* | *0.36* |
| **CR-10 at 800-m** | 8.90 | ± 0.7[a] | 6.40 | ± 1.2[b] | 5.30 | ± 0.9[c] | 4.60 | ± 1.0[c] | 9.50 | ± 0.5[a] | *p<0.0001* | *0.88* |

higher in 200m-1:1 than in 800m and 400m-2:1 sessions ($p<0.01$). At exhaustion, mean exercise HR, time spent in the 95–100% HR-zone and blood lactate differed for each session ($p<0.05$ in each parameter).

After the only accumulation of 800m, sessions' discrepancies appeared to increase for the exercise mean HR ($p = 0.01$), and 200m-1:1 presented a higher time spent in the 95–100% HR zone compared to othser sessions ($p<0.05$). Higher RPE were also recorded in the 800m and 400m-2:1 compared to the other sessions ($p<0.001$).

Fig 2 describes $TL_{total}$ according to sessions and methods. In most cases, quantifications by original methods appear similar. $TRIMP_c$ also appears to quantify TL in the same way as TRIMP, S-RPE and WER. TL for 400m-2:1 were lower than 400m-1:1 in TRIMP and $TRIMP_c$

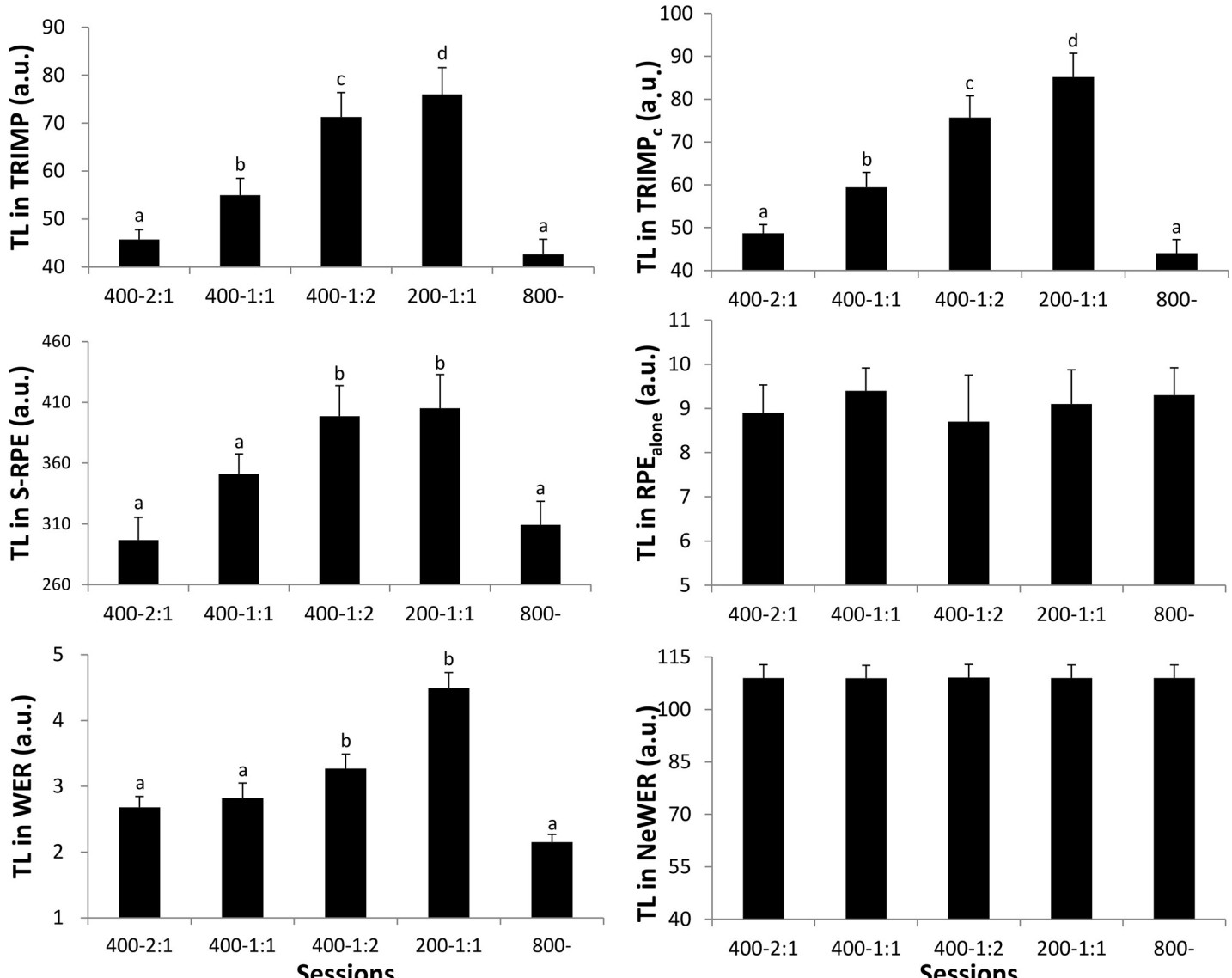

**Fig 2. Calculated training loads for the entire sessions using TRIMP and cumulative TRIMP methods, Session-RPE and RPE alone, Work Endurance Recovery (WER) and new WER.** Sessions comprised of warm-up and cool down periods and a 800m performance (800-), or of intervals of 400 or 200m (respectively 400 and 200m) with work: recovery ratios of 1:1, 2:1 and 1:2. In this figure, results with different manuscript letters (a, b, c, d, e) are significantly different from each other ($p<0.05$; one way-Anova and Tukey test) and those with the same letter are not. Data are presented as mean ± SD.

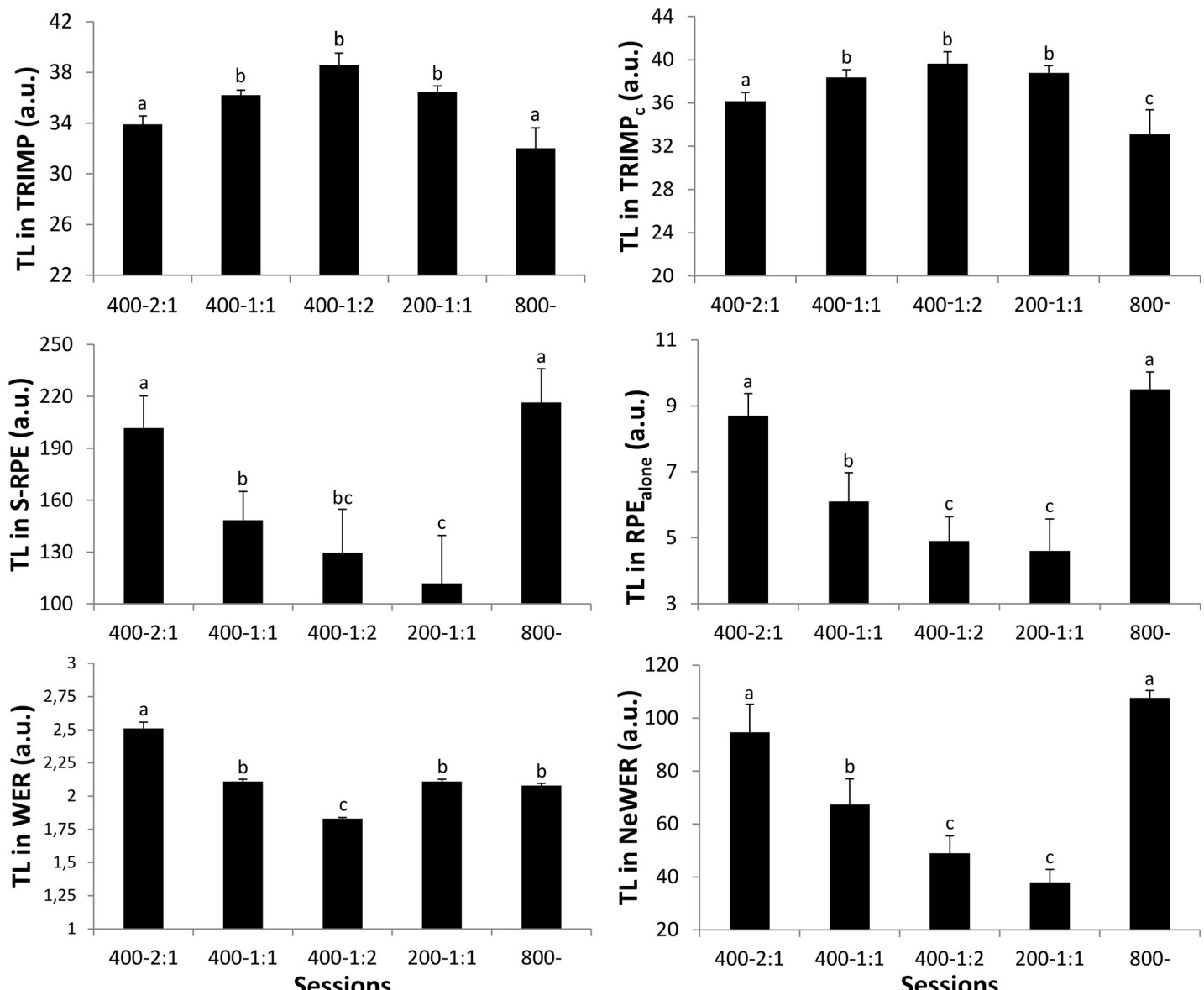

**Fig 3. Calculated training loads for 800m accumulated in the sessions using TRIMP and cumulative TRIMP methods, Session-RPE and RPE alone, Work Endurance Recovery (WER) and new WER.** Sessions comprised of a warm-up and a cool down periods and a 800-m performance (800-), or of intervals of 400 or 200 meters (respectively 400 and 200-) with work: recovery ratios of 1:1, 2:1 and 1:2. Results with different manuscript letters (a, b, c, d, e) are significantly different from each other (p<0.05; one way-Anova and Tukey test) and those with the same letter are not. Data are presented as mean ± SD.

(both, p<0.001) but not in S-RPE and WER (respectively, p = 0.1 and p = 0.1). TRIMP and TRIMPc report the highest TL for the 200m-1:1 session while this result did not differ in S-RPE and WER (respectively, p = 0.9 and p = 0.06). $TL_{total}$ for each exhausting session were equated when using $RPE_{alone}$ and NeWER (respectively, ANOVA with p = 0.2 and p = 0.5).

Fig 3 described $TL_{800}$ results from all methods. However TRIMP, unlike other methods, reported higher TL in 400m-1:1, 1:2 and 200m-1:1 compared to those of 400m-2:1 and 800m.

Table 2 reports the calculated correlations between methods for all TL ($TL_{total}$ and $TL_{800}$). High correlation levels appear in three original methods (TRIMP, S-RPE, WER), as well as in original methods and $TRIMP_c$. Two alternative methods ($RPE_{alone}$ and NeWER) are also

**Table 2. Matrix correlation of training loads for all sessions (total TL and TL when 800m of run accumulated) using 6 quantification methods (bottom of the matrix).** For correlation confidence intervals, see top of the matrix.

| *All* | $RPE_{alone}$ | WER | S-RPE | TRIMP | $TRIMP_c$ | NeWER |
|---|---|---|---|---|---|---|
| $RPE_{alone}$ | - | 0.26–0.57 | 0.69–0.84 | 0.16–0.51 | 0.19–0.53 | 0.85–0.93 |
| WER | 0.43 | - | 0.69–0.85 | 0.83–0.92 | 0.87–0.94 | 0.33–0.63 |
| S-RPE | 0.78 | 0.76 | - | 0.76–0.88 | 0.77–0.89 | 0.70–0.85 |
| TRIMP | 0.34 | 0.89 | 0.82 | - | 0.98–1.0 | 0.27–0.59 |
| $TRIMP_c$ | 0.34 | 0.91 | 0.82 | 0.99 | - | 0.29–0.60 |
| NeWER | 0.90 | 0.50 | 0.80 | 0.44 | 0.44 | - |

strongly correlated. $RPE_{alone}$ and NeWER are also correlated with S-RPE but not as strongly as with other original methods. When $TL_{total}$ only are considered, original methods and $TRIMP_c$ correlations remain strong (mean of r = 0.87 ± 0.08) whereas no significant correlation can be established from original or alternative methods, nor from $RPE_{alone}$ and NeWER (p>0.05). When only $TL_{800}$ are considered, TRIMP and $TRIMP_c$ are negatively correlated with the other methods (mean r = -0.69 ± 0.17). $RPE_{alone}$ is strongly correlated with NeWER and S-RPE (respectively, r = 0.92 and r = 0.99). WER is correlated with NeWER and S-RPE (respectively, r = 0.52 and r = 0.50).

## Discussion

To our knowledge, this is the first study that attempts to describe the marked differences between original quantification methods and their respective updated alternatives. TL quantification methods aim at describing the "dose" of effort that may subsequently be used to analyze the relations between the dose and the response to the dose [10]. TL quantification should only measure the exercise dose, which should be equal for all exercise constructions, without considering the effect of the dose. For accurate TL quantification, the causes for actual discrepancies between original and alternative methods must be clarified.

Decreases in interval distance and work:recovery ratios allow for the increase of the total exercise distance (800 in the 800m session *vs* 2690m in the 200m-1:1). The 800m session, and to a lesser extent the 400m-2:1, resulted in the highest values of exercise's HR and blood lactate concentrations which are frequently used as physiological parameters of exercise intensity [5,22]. Conversely, sessions with the less straining interval distance (200m-1:1) or work:recovery ratio (400m-1:2) resulted in longest times spent in the 95–100% HR zone. Finally, the distance covered in the 6min run of the cool-down period and RPE did not differ between the sessions, making it impossible to differentiate the sessions' induced fatigue [22].

For 800m accumulated runs, the discrepancies in physiological responses along the sessions were minored although RPE appeared higher in the 800m and 400m-2:1 sessions. The higher interval distances and work:recovery ratios the higher the RPE. The construction of exercise sessions differed, and so did the physiological factors inducing exhaustion, even though each session could induce a specific as well as maximal physiological strain.

$TL_{total}$ mainly showed great differences between $TRIMP_c$, original methods and $RPE_{alone}$ and NeWER. With $RPE_{alone}$ and NeWER, all the sessions performed until exhaustion resulted in similar TL measurements whereas for original methods the 800m and 400m-2:1 sessions resulted in lower TL measurements, ranging from 48% (WER) to 76% (S-RPE) of the values calculated for the 200m-1:1. $TL_{total}$ from original methods are strongly correlated but not with alternative methods. $TL_{total}$ from NeWER and $RPE_{alone}$ were not correlated due to NeWER's TL extremely low variations (CV = 3.5%) originating from slight TL changes of standardized warm-up and cool-down periods.

Some intriguing results appeared in $TL_{800}$. TRIMP and $TRIMP_c$ markedly differed from the other methods. After 800m accumulated runs, as the TRIMP methods showed, the lowest $TL_{800}$ report was in the 800m and 400m-2:1 sessions although exhaustion was nearly or even completely reached. This result could be explained by the fact that mean HR, largely impacted by warm-up periods, did not differ between sessions. TRIMP and $TRIMP_c$'s $TL_{800}$ therefore increased with exercise duration, which would vary according to the sessions' recovery periods and on the condition that mean HR for the sessions remained unchanged. In other words, the longer the recovery time the higher the TL.

Negative correlations between TRIMP methods and others contradict positive correlations reported in previous studies [4,10]. $TL_{800}$ in WER were also surprising. TL for the 800m session (where exhaustion was reached) was similar to that of the 400m-1:1 and 200m-1:1 sessions (respectively, exhaustion at 1360 and 2690m) and lower than in 400m-2:1. Conversely, the strong correlations between S-RPE and $RPE_{alone}$ seemed logical as the two methods were based on the same RPE value, and on the limited evolution of session volume used in S-RPE (ranging from $22.2 \pm 0.1$min to $26.5 \pm 0.3$). Although methods were moderately to strongly correlated with respect to $TL_{800}$, their relations still needed to be considered with care.

After all levels of quantifications were defined, strong correlations appeared in original methods (including $TRIMP_c$) as well as in alternative methods while low to moderate correlations were found when comparing alternative with original methods. Based on the discrepancies observed in TL quantification and moderate correlation levels, we assumed that original and alternative methods could not be interchangeable.

The opposition between original and alternative methods could be explained by the impact of exercise volume in quantification (S1 and S2 Appendices, S2 Fig). It is worth noting here that exercise volume in original methods was expressed in absolute values rather than relative to their maximum. Subsequently, intensity was multiplied by volume. Constructing methods in this way could be valid provided that the ranges of volume changes were equated to ranges of intensity. Yet, that was not the case. The higher TL values measured for longer sessions suggested that exercise volume had a higher impact on TL than exercise intensity or density. Considering TL as the physiological strain imposed on athletes, the higher TL provided by original methods in longer sessions suggested that such sessions induced higher physiological strain. Conversely, we assumed that exhaustion could be a means to detect maximal strain regardless of the sessions. In addition, we concluded that no single exercise parameter should prevail over others (*e.g.*, volume > intensity and density).

Moreover, the defense of original methods could be explained by the fact that, exercise volume, implying physiological adaptation to exercise, was deemed to have a major role to play on TL quantification. Furthermore, longer sessions could require longer recovery delays that might support their higher TL. However, responses to exercise and recovery delays were "effects" of the dose, which should have no impact on the "measurement" of the dose.

In practical terms, future applications should consider that original and alternative methods cannot be used similarly when exercises differ in volumes and density [10]. Considering that the mean session HR is a poor indicator of high intensity in interval exercises, the use of TRIMP methods should be limited to moderate intensity endurance exercises. In our results, RPE was markedly increased by the accumulation of runs (RPE at 800m *vs* at sessions' end) and by increased work:recovery ratios as observed after 800m accumulated runs. This demonstrates the impact of exercise's duration and density on ratings, supporting the use of $RPE_{alone}$ to quantify the exercise dose. When prescribing a training program, coaches and scientists may predict the exercise-induced RPE, or athletes may stop exercising once the expected RPE value is achieved. In this later case, RPE cannot be recorded 30min after the session's end as required in the S-RPE method. In addition, $RPE_{alone}$ is made to provide TL for the whole

session. However its relevance to quantify separately each exercise in any given session should be further investigated [23].

Although their components differ, NeWER and RPE$_{alone}$ are strongly correlated. NeWER is an objective method with which TL can be precisely pre-determined. NeWER suggests that TL may be assessed by the exercise-induced level of exhaustion in line with the "physiological strain" that TL should describe. Based on previous experience and/or more formal databases, coaches and scientist should determine which level of exhaustion they want to achieve with any given exercise. Further studies should help addressing this issue by providing tools for predicting exhaustion according to intermittent exercise's parameters.

As TL only measures the dose of exercise without considering its expected effect, training programs should not be solely based on TL but rather rely on adequate parameters and types of exercises. Program effectiveness after modifying TL or exercises must be tested through effects on performance, fatigue or physiological parameters.

Obviously, the present study was based on a small sample size and was conducted in a unique exercise type (intermittent) at one high intensity level. Therefore, our results should be supported by future studies focusing on more extended populations and using a wider range of exercises.

To conclude, significant differences in TL quantification appear between original and alternative methods stressing that they are not interchangeable. Conversely, due to their *modus operandi* and to the expression of exercise volume in absolute values, original methods may over-estimate TL in long duration sessions. Defining TL as the physiological strain imposed on athletes, it may be assumed that exhaustion is the prevalent metric to determine the highest physiological strain, whatever the exercise, and that exhaustion should match the highest possible TL. Therefore, this study mainly promotes the use of alternative methods for TL quantification which are based on exhaustion and perceived exertion.

## Supporting information

**S1 Fig. Examples of sample and theoretical quantiles plots for training quantification results.**
(TIF)

**S2 Fig. Relative contribution of original variables confronted to calculated training loads performed by principal analysis component.**
(TIF)

**S1 Appendix.**
(DOCX)

**S2 Appendix.**
(DOCX)

## Acknowledgments

We want to thank Philippe Bardy and Aurélie Brun for their careful reviews of the article.

## Author Contributions

**Conceptualization:** François-Denis Desgorces, Jean-François Toussaint, Philippe Noirez.

**Formal analysis:** François-Denis Desgorces, Jean-Christophe Hourcade, Romain Dubois, Philippe Noirez.

**Funding acquisition:** Jean-François Toussaint.

**Investigation:** François-Denis Desgorces, Jean-Christophe Hourcade, Romain Dubois.

**Methodology:** François-Denis Desgorces, Jean-Christophe Hourcade, Philippe Noirez.

**Project administration:** Jean-François Toussaint.

**Supervision:** François-Denis Desgorces.

**Visualization:** Philippe Noirez.

**Writing – original draft:** François-Denis Desgorces.

**Writing – review & editing:** Romain Dubois, Jean-François Toussaint, Philippe Noirez.

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
