## [Decision Letter · Decision Letter 0]

14 Apr 2020

PONE-D-20-08508

Training load quantification of high intensity exercises: discrepancies between classical and alternative methods

PLOS ONE

Dear Dr DESGORCES,

Thank you for submitting your manuscript to PLOS ONE. After careful consideration, we feel that it has merit but does not fully meet PLOS ONE’s publication criteria as it currently stands. Therefore, we invite you to submit a revised version of the manuscript that addresses the points raised during the review process.

Among other, reviewers have been raised statistical / methodological weaknesses, have underlined that clarity and flow could be improved both, together with English level.

We would appreciate receiving your revised manuscript by May 29 2020 11:59PM. To enhance the reproducibility of your results, we recommend that if applicable you deposit your laboratory protocols in protocols.io, where a protocol can be assigned its own identifier (DOI) such that it can be cited independently in the future. For instructions see: http://journals.plos.org/plosone/s/submission-guidelines#loc-laboratory-protocols

We look forward to receiving your revised manuscript.

Kind regards,

Laurent Mourot

Academic Editor

PLOS ONE

Journal Requirements:

"NO. The funders had no role in study design, data collection and analysis, decision to publish, or preparation of the manuscript."

Reviewers' comments:

Reviewer's Responses to Questions

**Comments to the Author**

1. Is the manuscript technically sound, and do the data support the conclusions?

Reviewer #1: Partly

Reviewer #2: Partly

Reviewer #3: Yes

2. Has the statistical analysis been performed appropriately and rigorously? 

Reviewer #1: Yes

Reviewer #2: Yes

Reviewer #3: No

3. Have the authors made all data underlying the findings in their manuscript fully available?

Reviewer #1: Yes

Reviewer #2: Yes

Reviewer #3: No

4. Is the manuscript presented in an intelligible fashion and written in standard English?

Reviewer #1: No

Reviewer #2: Yes

Reviewer #3: Yes

5. Review Comments to the Author

Reviewer #1: The manuscript titled, “Training load quantification of high intensity exercises: discrepancies between classical and alternative methods,” presented a research study with the main aim of comparing classical and alternative methods of quantifying training load during high intensity exercises. The mode of exercise included five sessions of outdoor running on a track: 1) 800-m maximal effort; 2) 400-m intervals until exhaustion with a 2:1 work to recovery ratio; 3) 400-m intervals until exhaustion with a 1:1 work to recovery ratio; 4) 400-m intervals until exhaustion with a 1:2 work to recovery ratio; and 5) 200-m intervals until exhaustion with a 1:1 work to recovery ratio. The topic is of practical importance and can help guide athletes, coaches, and sport scientists on the best methods to quantify training load for high intensity sessions such as track interval work. The authors main findings were that the training load calculations are not interchangeable. Their recommendation is that when the exercise session involves exhaustion, alternative training load calculations should be used that account for exhaustion and perceived exertion in their calculations.

Major Comments:

1) The manuscript includes a number of grammatical errors. I would suggest the authors address this to the best of their ability. This will improve clarity as well as sentence structure.

2) In your discussion of training load, it would be helpful to identify and define internal and external training load. The dose of exercise is the external training load (800-m sprint for example). The response to the dose of exercise is the internal training load (RPE for example).

3) There is not much literature on the work endurance recovery formula. The authors should justify why they believe this is a classical method. Please provide more background on where it came from and reasoning for creating an alternative formula to the work endurance recovery formula. My understanding from reading reference #3 is that the calculation provides a more accurate training load quantification when a training session included intermittent exercise such as a strength training session.

4) It is unclear whether the work to recovery ratio was time or distance. For example, in the 400m 2:1 ratio, did the participants run the 400-m interval and then walk-jog for 200m recovery or did they have a rest period of half the duration of the 400-m interval? Please clarify this.

5) The authors defined TL800 as a distance of 800-m at specific velocity. It is unclear what portion of the workout this 800-m is taken from. From Table 1 it appears some of the 800-m segment includes recovery distances. I believe it would be more relevant to compare 800-m from each workout at the 800-m race velocity.

6) Is it the authors’ position that training load for each workout should be similar because each workout achieved maximal physiological strain (done to exhaustion)? I believe the benefit of rest periods in interval sessions is that the athlete can perform more work overall. Thus the training load for the 400m 1:2 and 200m 1:1 should be higher. The athlete was able to run farther at 800-m velocity. This additional work should be taken into account. Stephen Seiler’s work compares a fixed amount of work across a range of interval prescriptions. This is what I thought TL800 was meant to be, but the data indicates TL800 includes recovery periods.

7) I believe if the authors presented the findings in a more unbiased way, it would strengthen the paper. In addition, the paper would be greatly strengthened if the paper had more clarity regarding methodology. A figure showing the segment of TL800 would help.

I will reserve a majority of my minor comments with the revision. However a few are listed below:

1) It looks neater when the formulas are separated out from the main text.

2) Please indicate how you defined exhaustion (when participants stopped the interval session).

3) Please clarify what you mean by density of exercise.

4) Did the researchers record temperature, wind speed, and/or humidity for the training sessions? Please indicate.

Reviewer #2: The paper " Training load quantification of high intensity exercises: discrepancies between classical and alternative methods" is very interesting. This study is in a relevant area and be should be replied with other type of participants or sports. The authors should be commended for their hard work in what appears to be an extensive study. In my point of view, the current form of the manuscript does not provide practical impact in research field, but it may provide if some further details were given. Hence, I've recommended revision to improve further text clarity before I can consider recommending it for publishing.

Abstract

L 27-29 – I believe that would be better if you simplify the sentence to “The purpose of the present study was to quantify training loads (TL) by classical methods (TRIMP; session-RPE; work-endurance-recovery) and their alternatives (TRIMPcumulative; RPEalone; New WER).”

L 29 – The author should avoid beginning sentences with numbers.

L 45-46 – I don’t really understand what you mean. You should focus on presenting your conclusions.

L46-48 – Although I understand that authors want to make some suggestions for future research, I would prefer to read some main conclusions in the last lines of the abstract. Otherwise, a reader will keep the idea that.

Manuscript

L52 – It should be “forty-five years ago”.

L67-68 – There a lot of scientific papers that says the opposite. I would be careful with sentences like this.

L171-172 – I not sure If I understand what you mean. You assumed that HRmax was the highest HR reached during 800-m running? If so, I do not agree with this procedure. Did you consider accessing HRmax by applicating a maximum cardiorespiratory test? How can you confirm that HRmax achieved in the 800-m running was the HRmax that a participant can achieve?

L183-186 – you recorded RPE 30min following the end of the session, but you also recorded immediately after the 800-m performance. How do you use both values? I suggest a better clarification.

L225 – it is not clear what does mean letters a, b, c, d and e. they are significantly different from…?

L239-240 and 247-253 – The same comment regarding figures 1 and 2.

Also, it is recommended to include SD in the figures presented.

L277-278 – This is speculative. Can you point what were the previous analysis that you are referring?

L302 – what do you mean by achieving exhaustion? Is that mean that a participant can no longer maintain their exercise intensity?

L303-304 – I want to ask if the authors can provide the analysis with including warm-up and cool down session. Despite the majority of the studies regarding TL quantification include those parts of the session, it would be interesting to analyse the differences between the methods used without including warm-up and cool down.

L322-323 – That assumption seems obvious. In order to clarify, can you provide some guidelines for future use or how to choose a method to TL quantification? Can you add in Practical applications section?

L324-333 – I don’t know how that paragraph contributes with new knowledge. There are a lot of studies that analysed the construct and validity of RPE or S-RPE that already state what you mention. I know you are trying to build a text with logical following, but can you add any new advice regarding this topic?

L339 – “…may be uneasy and inefficient for determining TL of exercises or programs to come”. Can you clarify what you mean? Because I can enumerate a lot of studies that support the use of RPE.

L352 – I think there are something missing in the end of the sentence, once you put “()” with no information on it.

L372 – I suggest that you specify some methods by giving some examples.

In addition, there no Practical applications or limitations presented. Those sections should be reviewed. For instance, after reading, a reader still does not know what TL quantification method should use. Also, some limitations, e.g. the small sample size should be noticed.

Reviewer #3: 1. Is the manuscript technically sound, and do the data support the conclusions?

The purpose of the study is of interest in the field of training and athletic preparation. Unfortunately, to my opinion, the cohort is very small and do not allow generalization of the results obtained, especially based on the statistical procedure used. Authors should state this at the end of the discussion section. The protocol is well conducted and supports the replication of the study.

2. Has the statistical analysis been performed appropriately and rigorously?

To my opinion, the statistical analysis which has been performed is the main weakness of the study. Based on 10 athletes (which is a very small cohort), the authors tested the normality of the data using a Kolmogorov-Smirnov test. According to the state of my knowledge, for small samples, it is currently recommended to use the Shapiro–Wilk test. Indeed, a test of significance for normality may lack power to detect the deviation of the variable from normality. Shapiro–Wilk test was designed to test for normality for small data-size (n < 50). This test is more powerful than Lillifors or Kolmogorov-Smirnov and other tests for small data-size. Then I recommend making a Shapiro–Wilk test to assess the normality of the data.

Based on the Kolmogorov-Smirnov test, authors have made one-way ANOVA and Pearson’s correlations. Using ANOVA needs that the residuals respect the normality assumption. In this line, and considering the small number of subjects, I recommend putting the QQ-plot in the supplementary materials so that others can see what authors judged to be normal, regardless of what their normality test says about residuals. I think it is important because such things are tricky with small sample sizes.

Another matter considering the results obtained in this study is the power of the tests. In this line, I recommend two things. Firstly, please provide power analysis to estimate if the sample size is large enough for the protocol and statistical analysis used. Secondly, add in the results section effect size and/or confident intervals (ANOVA, correlations) to support the results and to allow discussions.

3. Have the authors made all data underlying the findings in their manuscript fully available?

Authors report that some restrictions will apply and that all relevant data are within the manuscript and its Supporting Information files.

4. Is the manuscript presented in an intelligible fashion and written in standard English?

Line 55. TL is defined as the exercise-induced physiological strain resulting from the combination of exercise’s intensity, volume and density influences.

Please add a rationale for using the RPE scale which includes a subjective aspect in the training load evaluation.

Line 65. As example, pharmaceutics weight is linked to their weighing and not to their expected effects.

Please, provide a clearer example, or perhaps elaborate on the argument.

L 100. They were in good mental and physical condition, and were not under any medication.

Please, provide information concerning how these aspects were controlled.

L 109. Each session was performed at 10.00 a.m. on an outdoor athletics track with similar climatic conditions. Subjects were asked to arrive for each session in a rested state, suspend their normal training for at least 24h prior to session and avoid exhausting exercises in the study period.

This means changing the usual athletes’ training program. In your opinion, what would have been the consequences of the prior charges on the states? Moreover, what is the point of doing that since we are going to try to measure a load induced by specific exercises? Shouldn't the effect you observed be independent of the initial form state? The fitness standardization is commendable but wouldn't it have been better to have a population carrying out the same training program during the 3 weeks of the study? Please, give a rationale for this standardization.

L 121. interspersed by at least three days.

Please, give Mean/SD.

L 172. HRb was self-measured when the subjects awoke in three consecutive mornings, the lower value measured was recorded.

Please, provide information on the process that has been followed. Anatomical position and recording duration.

L 179. RPE was obtained using the category ratio scale (CR-10) modified by [15].

1. Please argue why you used this tool instead of the original one.

2. Rewrite [15]

L 330. Therefore, RPE may reflect exercise intensity but also relative volume and density, suggesting that, in S-RPE method, exercise volume is accounted twice (i.e., in RPE and in volume itself). Conversely, RPE itself could be used to monitor and assess, at an individual basis, the exercise-induced physiological strain that is the definition of TL.

To my knowledge, the session-duration refers to the length of session expressed in minutes while the nominal score is given by an athlete to describe his RPE of “mean training intensity” during that training session. In this perspective, there is no reason to delete the time volume from the calculation. Please add a more consistent rationale supporting this sentence.

L 352. than exercise intensity or density ().

Please, delete brackets or add items.

6. PLOS authors have the option to publish the peer review history of their article (what does this mean?). If published, this will include your full peer review and any attached files.

Reviewer #1: No

Reviewer #2: Yes: Rafael Franco Soares Oliveira

Reviewer #3: No

---

## [Author Response · Author response to Decision Letter 0]

26 Jun 2020

Responses have been uploaded with attached files.

---

## [Decision Letter · Decision Letter 1]

20 Jul 2020

Training load quantification of high intensity exercises: discrepancies between original and alternative methods

PONE-D-20-08508R1

Dear Dr. DESGORCES,

We’re pleased to inform you that your manuscript has been judged scientifically suitable for publication and will be formally accepted for publication once it meets all outstanding technical requirements.

Kind regards,

Laurent Mourot

Academic Editor

PLOS ONE

Additional Editor Comments (optional):

Reviewers' comments:

Reviewer's Responses to Questions

**Comments to the Author**

1. If the authors have adequately addressed your comments raised in a previous round of review and you feel that this manuscript is now acceptable for publication, you may indicate that here to bypass the “Comments to the Author” section, enter your conflict of interest statement in the “Confidential to Editor” section, and submit your "Accept" recommendation.

Reviewer #2: All comments have been addressed

Reviewer #3: All comments have been addressed

2. Is the manuscript technically sound, and do the data support the conclusions?

Reviewer #2: Yes

Reviewer #3: (No Response)

3. Has the statistical analysis been performed appropriately and rigorously? 

Reviewer #2: Yes

Reviewer #3: (No Response)

4. Have the authors made all data underlying the findings in their manuscript fully available?

Reviewer #2: Yes

Reviewer #3: (No Response)

5. Is the manuscript presented in an intelligible fashion and written in standard English?

Reviewer #2: Yes

Reviewer #3: (No Response)

6. Review Comments to the Author

Reviewer #2: The paper "Training load quantification of high intensity exercises: discrepancies between original and alternative methods " is a good contribution for the current state of the art in this specific topic. This study highlights more knowledge on training load quantification methods that, per se, are very useful in different sports, physical activities and/or exercise training programs. They allow a better training control for different athletes or non-athletes. Therefore, the authors should be commended for their hard work in what appears to be an extensive study. Now, the current form of the manuscript provides practical applications in this field that can be applied in other similar studies or other contexts. With the improvements made by the authors, the study can be easily replied which is very important for scientific community in sports science. Now, I would like to congratulate the authors for this revised version of the manuscript, as I now recommend it to be accept. Although, I would like to suggest to the authors add some references or context in the introduction regarding blood lactate use or its advantage use. Also, it would be better, if the authors clarify what the letters a, b, c, d and e mean in figures 2 and 3 (probably some differences between the different exercises).

Reviewer #3: (No Response)

7. PLOS authors have the option to publish the peer review history of their article (what does this mean?). If published, this will include your full peer review and any attached files.

Reviewer #2: **Yes: **Rafael Franco Soares Oliveira

Reviewer #3: No

---

## [Editor Report · Acceptance letter]

23 Jul 2020

PONE-D-20-08508R1 

Training load quantification of high intensity exercises: discrepancies between original and alternative methods 

Dear Dr. Desgorces:

I'm pleased to inform you that your manuscript has been deemed suitable for publication in PLOS ONE. Congratulations! Your manuscript is now with our production department. 

Kind regards, 

on behalf of

Dr Laurent Mourot 

Academic Editor

PLOS ONE